# Association between prehospital time and outcome of trauma patients in 4 Asian countries: A cross-national, multicenter cohort study

Chi-Hsin Chen[1], Sang Do Shin[2], Jen-Tang Sun[3], Sabariah Faizah Jamaluddin[4], Hideharu Tanaka[5], Kyoung Jun Song[2], Kentaro Kajino[6], Akio Kimura[7], Edward Pei-Chuan Huang[1,8], Ming-Ju Hsieh[8], Matthew Huei-Ming Ma[8,9], Wen-Chu Chiang[8,9]*

1 Department of Emergency Medicine, National Taiwan University Hospital Hsin-Chu Branch, Hsin-Chu City, Taiwan, 2 Department of Emergency Medicine, Seoul National University College of Medicine and Hospital, Seoul, Korea, 3 Department of Emergency Medicine, Far Eastern Memorial Hospital, New Taipei City, Taiwan, 4 Faculty of Medicine, Universiti Teknologi MARA, Shah Alam, Malaysia, 5 Graduate School of Emergency Medical Service System, Kokushikan University, Tokyo, Japan, 6 Traumatology and Critical Care Medical Center, National Hospital Organization Osaka National Hospital, Osaka, Japan, 7 Department of Emergency Medicine and Critical Care, Center Hospital of the National Center for Global Health and Medicine, Tokyo, Japan, 8 Department of Emergency Medicine, National Taiwan University Hospital, Taipei City, Taiwan, 9 Department of Emergency Medicine, National Taiwan University Hospital Yun-Lin Branch, Douliu City, Taiwan

* drchiang.tw@gmail.com

**Data Availability Statement:** The data of this study are owned by the PATOS coordination center. The

## Abstract

### Background

Whether rapid transportation can benefit patients with trauma remains controversial. We determined the association between prehospital time and outcome to explore the concept of the "golden hour" for injured patients.

### Methods and findings

We conducted a retrospective cohort study of trauma patients transported from the scene to hospitals by emergency medical service (EMS) from January 1, 2016, to November 30, 2018, using data from the Pan-Asia Trauma Outcomes Study (PATOS) database. Prehospital time intervals were categorized into response time (RT), scene to hospital time (SH), and total prehospital time (TPT). The outcomes were 30-day mortality and functional status at hospital discharge. Multivariable logistic regression was used to investigate the association of prehospital time and outcomes to adjust for factors including age, sex, mechanism and type of injury, Injury Severity Score (ISS), Revised Trauma Score (RTS), and prehospital interventions. Overall, 24,365 patients from 4 countries (645 patients from Japan, 16,476 patients from Korea, 5,358 patients from Malaysia, and 1,886 patients from Taiwan) were included in the analysis. Among included patients, the median age was 45 years (lower quartile [Q1]–upper quartile [Q3]: 25–62), and 15,498 (63.6%) patients were male. Median (Q1–Q3) RT, SH, and TPT were 20 (Q1–Q3: 12–39), 21 (Q1–Q3: 16–29), and 47 (Q1–Q3:

data were not freely available because of the regulations of PATOS organizations, but an application to get the data is possible, if the application is approved by the PATOS EXCO meeting mainly composing of data-contributing principal investigators from Asia countries. The contact of the PATOS coordination center is listed below. Ms. Suhee Agnes KIM, MPH. Laboratory of Emergency Medical Services, Biomedical Research Institute. Seoul National University Hospital, Seoul, South Korea. Tel: +2 2072 4683; Mobile: +82 10 8572 7715; http://lems.re.kr/ Email: suheekimsnuh@gmail.com

**Funding:** This study was funded by the Taiwan Ministry of Science and Technology (MOST 108-2314-B-002-130-MY3 and MOST 105-2314-B-002-200-MY3 and MOST 109-2314-B-002 -154 -MY2). The funders had no role in study design, data collection and analysis, decision to publish, or preparation of the manuscript.

**Competing interests:** The authors have declared that no competing interests exist.

**Abbreviations:** AIS, Abbreviated Injury Scale; aOR, adjusted odds ratio; AUROC, area under the receiver operating characteristic curve; ED, emergency department; EMS, emergency medical service; EMT, emergency medicine technician; IO, intraosseous line; ISS, Injury Severity Score; IV, intravenous line; MRS, Modified Rankin Scale; PATOS, Pan-Asia Trauma Outcomes Study; RT, response time; RTS, Revised Trauma Score; SH, scene to hospital time; TBI, traumatic brain injury; TPT, total prehospital time.

32–60) minutes, respectively. In all, 280 patients (1.1%) died within 30 days after injury. Prehospital time intervals were not associated with 30-day mortality. The adjusted odds ratios (aORs) per 10 minutes of RT, SH, and TPT were 0.99 (95% CI 0.92–1.06, $p = 0.740$), 1.08 (95% CI 1.00–1.17, $p = 0.065$), and 1.03 (95% CI 0.98–1.09, $p = 0.236$), respectively. However, long prehospital time was detrimental to functional survival. The aORs of RT, SH, and TPT per 10-minute delay were 1.06 (95% CI 1.04–1.08, $p < 0.001$), 1.05 (95% CI 1.01–1.08, $p = 0.007$), and 1.06 (95% CI 1.04–1.08, $p < 0.001$), respectively. The key limitation of our study is the missing data inherent to the retrospective design. Another major limitation is the aggregate nature of the data from different countries and unaccounted confounders such as in-hospital management.

## Conclusions

Longer prehospital time was not associated with an increased risk of 30-day mortality, but it may be associated with increased risk of poor functional outcomes in injured patients. This finding supports the concept of the "golden hour" for trauma patients during prehospital care in the countries studied.

---

## Author summary

### Why was this study done?

- The concept of the "golden hour from injury to definitive care," suggesting that critically injured patients should receive definite treatment in 60 minutes, was first proposed early in the 20th century and has been challenged because studies have shown divergence in the association between prehospital time and mortality in injured patients.

- To our knowledge, there has never been a study to adapt functional status as an outcome measurement for the impact of prehospital time in injured patients.

### What did the researchers do and find?

- This 3-year, cross-national, multi-center cohort study included 24,365 patients from 4 Asian countries (Japan, Korea, Malaysia, and Taiwan).

- We found no association between prehospital time and 30-day mortality in trauma patients overall, but longer prehospital time was detrimental to functional outcome. Every 10-minute delay in total prehospital time was associated with a 6% increase in the odds of a poor functional outcome. Poor functional outcome indicates severe disability in daily life, or death.

### What do these findings mean?

- Trauma patients who experienced prehospital delays were likely to have poorer functional outcomes in the countries studied.

- The prehospital delays may arise from the response time, scene control, extrication, interventions, and transportation in the prehospital setting. These findings remind the prehospital staff to optimize the prehospital time to promote favorable functional outcomes for trauma patients.

- Our analysis is susceptible to potential bias resulting from the aggregate nature of the data from different countries, unaccounted confounders such as quality of prehospital care and in-hospital management, and missing data inherent to the retrospective design.

- Policymakers from different countries and areas should make an effort to examine the influence of prehospital time and to develop suitable prehospital guidelines based on their own emergency medical service configurations.

## Introduction

Trauma is one of the leading causes of death and contributes to approximately 0.5% of the mortality annually worldwide [1,2]. Prehospital emergency medical service (EMS) can provide timely resuscitation and transportation of critically injured patients to medical care facilities and, therefore, is crucial in trauma care [3]. Prehospital time, consisting of response time, on-scene time, and transport time, is an essential parameter of EMS, which may potentially influence trauma patients' outcomes. Many factors may have a potential impact on prehospital time, including the response speed of the EMS; distance between the scene, the local EMS department, and medical facilities; multiple environmental factors; severity of condition; and on-scene management [4]. Although there have been significant advances in resuscitation, airway management, circulatory access, and hemorrhage control as the prehospital care provided by paramedics and other emergency medical technicians, on-scene interventions can increase prehospital time [3].

Whether reducing prehospital time by rapid transportation can reduce mortality remains debated [5]. The concept of the "golden hour" was first proposed in the 20th century, suggesting that critically injured patients should receive definite treatment within 60 minutes [6]. Many studies have been performed to determine the association between prehospital time and outcome, showing inconsistent results [7–14]. Most of them showed no significant correlation, or even that shorter prehospital time was correlated with poorer outcomes [7,8]. Owing to the lack of clear evidence on the effect of shortening prehospital time in patients with trauma, several articles aimed to identify specific injury subgroups that may benefit from reduced prehospital time, such as younger patients or patients with traumatic brain injury (TBI) [9,10]. Some studies showed a benefit of rapid response or transportation in cases of penetrating injury [11–13]. A systematic review was conducted by Harmsen et al., who concluded that rapid transportation may be beneficial for patients with neurotrauma and penetrating injury with unstable hemodynamic features [14].

Aside from the inconsistencies in the findings of previous studies mentioned above, only 1 study to our knowledge has involved an Asian population; this study focused on the effect of prehospital time in regions without prehospital care in India [15]. Asian countries have different EMS systems, trauma care provisions, and distributions of rural and urban areas compared to Western countries. Demographic differences are also prominent among Asian countries. A

few developed countries in Asia have helicopter transportation of injured patients to tertiary medical centers and trauma specialists, while other Asian countries still have incomplete trauma care systems [16]. Therefore, specific studies involving Asian populations are needed to better understand the Asian situation and to comparatively analyze the situation in other continents in the world.

Our study aimed to determine the impact of prehospital time on patient outcomes, whether rapid transportation can benefit trauma patients, and the impact of the "golden hour" on injured patients in Asian populations. We hypothesized that trauma patients would benefit from rapid transportation, and that the concept of golden hour would be applicable in prehospital care for trauma in Asia. Additionally, we tried to investigate potential factors that may influence prehospital time. If the result showed that longer prehospital time is detrimental to patients' outcomes, the findings of this study could further guide the EMS in shortening prehospital time.

## Methods

### Study design and setting

We conducted a retrospective cohort study of trauma patients admitted to the emergency department (ED) in the included countries from January 1, 2016, to November 30, 2018. This study is reported as per the Strengthening the Reporting of Observational Studies in Epidemiology (STROBE) guideline (S1 Checklist) [17].

Patient data were retrospectively reviewed from the Pan-Asia Trauma Outcomes Study (PATOS), which was initiated in 2015, with data on the following variables: epidemiologic factors, EMS, ED care, hospital care and management factors, and records of final outcomes. This cross-national trauma registry consisted of 33 sites in 14 Asian countries such as China, India, Korea, Japan, Malaysia, Thailand, the Philippines, Taiwan, and Vietnam [18,19]. Most of the EMS and trauma care systems involved in this registry were in the urban cities of each country [19]. Participation in the PATOS registry is voluntary. Patients' data were recorded in the registry if they were sent to the participating hospitals due to trauma, either from the scene or via inter-hospital transport. The only exception was data from Taiwan, where the registry data had information on EMS-transported trauma patients who met the criteria of prehospital activation for major trauma.

### Selection of participants

Patients were included if they were transported from the scene to hospital by the EMS during the period from January 1, 2016, to November 30, 2018. The exclusion criterion for the primary study cohort was missing data on age, sex, 30-day mortality, Revised Trauma Score (RTS) at the ED, Injury Severity Score (ISS), or prehospital time interval (response time [RT], scene to hospital time [SH], and total prehospital time [TPT]). In the secondary cohort, we further excluded patients with missing data on functional outcome at discharge.

### Ethics statement

The PATOS collaboration was approved by the institutional review board of the National Taiwan University Hospital. The data were fully anonymized at the time they were accessed by the authors. The study proposal and analysis plan were approved at the PATOS Taipei meeting, on November 7, 2015. The original study proposal is shown in Table A in S1 Text. The PATOS trauma database is characterized as an EMS-based registry. The initial proposal (2015) of the current study planned to analyze the association between "the golden hour (i.e. injury

time to definite care in hospital)" and trauma patient outcomes. However, when the PATOS phase 1 data were released in 2019, we found that there was an inadequate sample size of patients with ISS ≥ 16; therefore, the analysis plan was revised to test the association between prehospital timeliness and outcome.

## Measurements

**Variables.** The basic characteristics of the patients included in our study included country, age, sex, mechanism of injury (penetrating or non-penetrating injury), and type of injury (non-TBI, mixed TBI, or isolated TBI). Patients were divided into non-geriatric patients (age < 60 years) and geriatric patients (age ≥ 60 years) in the subgroup analysis [20]. Data on prehospital management such as rescue airway (supraglottic airway or endotracheal tube) and the establishment of fluid access either by intravenous line (IV) or by intraosseous line (IO) were collected. We adopted the ISS and RTS as the indices of trauma severity. ISS was calculated by summing the square of the 3 highest Abbreviated Injury Scale (AIS) scores for injuries to different body regions [21]. Among the participating countries considered in the final analysis, Korea, Japan, and Malaysia reported AIS 2008 codes (7 digits) to the PATOS database, whereas Taiwan used AIS 1995 codes (5 digits) but manually converted them to the approximate AIS 2008 codes. RTS, a physiological triage score, was calculated using recoded Glasgow Coma Scale (GCS), systolic blood pressure (SBP), and respiratory rate (RR) using the following formula: RTS = (GCS score coded × 0.9368) + (SBP coded × 0.7326) + (RR coded × 0.2908) [22]. We further dichotomized ISS as <16 or ≥16 and RTS as <7 or ≥7; these cutoff values were used in previous studies as cutoff values for major trauma [23–25]. The prehospital time intervals were extracted using prehospital timing with maximal valid value (Table B in S1 Text). For the prehospital time intervals in the PATOS database, one of the major contributing countries (Korea) mostly reported the SH instead of transport time in their uploaded data. We could not exactly estimate the transport time using this part of the Korean data, and excluding all of this information would harm the statistical power required to test the hypothesis. Hence, the time intervals used in our study were RT (from the time of injury to the time of EMS arrival at the scene), SH (from the time of EMS arrival at the scene to the time of EMS arrival at the ED), and TPT (from the time of injury to the time of EMS arrival at the ED). After calculating the above variables, data were replaced as missing values if RT was lower than 0 minutes or higher than 120 minutes, SH was lower than 0 minutes or higher than 240 minutes, TPT was lower than 0 minutes or higher than 360 minutes, or age was below 0 years or above 120 years, which were considered unreasonable data and potential outliers. Additionally, we further compared the basic characteristics of the included sample and the sample excluded due to missing data, in response to peer review.

**Outcomes.** The primary outcome was mortality within 30 days after injury, which is a standard follow-up period for major trauma based on the Utstein template [26]. In response to peer review, we also performed a sensitivity analysis on different outcomes: mortality within 24 hours, 14 days, 30 days, or 60 days. Another outcome measurement used in our study was functional status at discharge based on the Modified Rankin Scale (MRS). The MRS data from Korea and Japan were obtained by the physician (usually a resident or intern) with a structured questionnaire, while in Malaysia, data were collected by the nursing staff during discharge without a structured questionnaire. Data from Taiwan did not evaluate MRS during the period 2016 to 2018. MRS was initially used as an index of functional outcome in patients with stroke [27]. The scale was further applied to measure the disability caused by TBI or general trauma [28–31]. No significant disability, slight disability, or moderate disability (MRS 0–3) were defined as favorable functional outcomes, and moderately severe disability, severe disability, and death (MRS 4–6) were categorized as poor functional outcomes [32].

## Statistical analysis

All continuous data that were not normally distributed were subjected to the Kolmogorov–Smirnov test. Dichotomous and categorical variables are reported as absolute sample size (percentage), whereas continuous variables are reported as median (lower quartile [Q1]–upper quartile [Q3]). Continuous variables were compared using non-parametric ANOVA or Mann–Whitney U test. Categorical and nominal variables were compared using Pearson chi-squared test or Fisher's exact test.

Multivariable logistic regression was used to determine the association between prehospital time and mortality in trauma patients within 30 days and functional outcome at discharge. Variables that had $p < 0.10$ on the chi-squared test or Mann–Whitney U test were selected for multivariable logistic regression analysis using the forced entry method. The discrimination of the regression model was tested using the area under the receiver operating characteristic curve (AUROC) for each outcome. The model fit was assessed using the Hosmer–Lemeshow goodness-of-fit test [33]. In response to peer review, we also depicted plots of predicted to observed outcome rates of TPT per 10 minutes to permit visual inspection of model calibration [34,35]. The association of prehospital time and predicted possibility of favorable functional outcome was determined using restricted cubic spline regression. To determine the potential factors that may influence TPT, variables that had $p < 0.10$ on the Mann–Whitney U test were selected for multivariable linear regression analysis, as the dependent variable is continuous, using the forced entry method. The receiver operating characteristic curve and Youden Index (YI) were used to determine the best cutoff value of prehospital time to predict the outcomes [36]. Statistical analysis was performed using SPSS version 26.0 (IBM, Armonk, NY, US). All tests were 2-sided, and a $p$-value $< 0.05$ was considered statistically significant.

## Results

### Characteristics of study objects

A total of 71,383 patients were eligible for review in the database. After exclusion of patients who underwent inter-hospital transport or were not transported by the EMS ($n = 21,494$) or whose date of injury was outside the study period or unknown ($n = 1,541$), 48,348 patients remained. Among them, 23,909 patients were excluded for missing data on age, sex, mortality at 30 days, ISS, RTS, RT, SH, or TPT. Records of 2 countries (Thailand and Vietnam) with an insufficient number of included individuals were also excluded ($n = 74$). The remaining 24,365 patients were included in the study of 30-day mortality. In this study cohort for the primary outcome, 21,886 patients had a valid record of MRS at discharge and were therefore further included in the study of functional outcome. A detailed flow diagram is presented in Fig 1.

The demographics of the 24,365 patients included in the study of 30-day mortality are detailed in Table 1. This study cohort consisted of patients from 4 countries, including Japan, Korea, Malaysia, and Taiwan. All hospitals from these countries participating in the investigation are academic teaching hospitals with functional capability for trauma resuscitation. The median (Q1–Q3) of RT, SH, and TPT were 20 (Q1–Q3: 12–39), 21 (Q1–Q3: 16–29), and 47 (Q1–Q3: 32–60) minutes. Mortality within 30 days after injury occurred in 280 patients (1.1%). Thirty-day mortality was significantly higher in older patients (median age 61.0 versus 45.0 years, $p < 0.001$) and in those with non-penetrating injury (99.3% versus 95.8%, $p = 0.004$), TBI (isolated TBI, 33.6%; mixed-TBI, 40.0%; non-TBI, 26.4%; $p < 0.001$), and major trauma (ISS $\geq$ 16, 66.1% versus 6.0%, $p < 0.001$; RTS $< 7$, 74.6% versus 4.0%, $p < 0.001$).

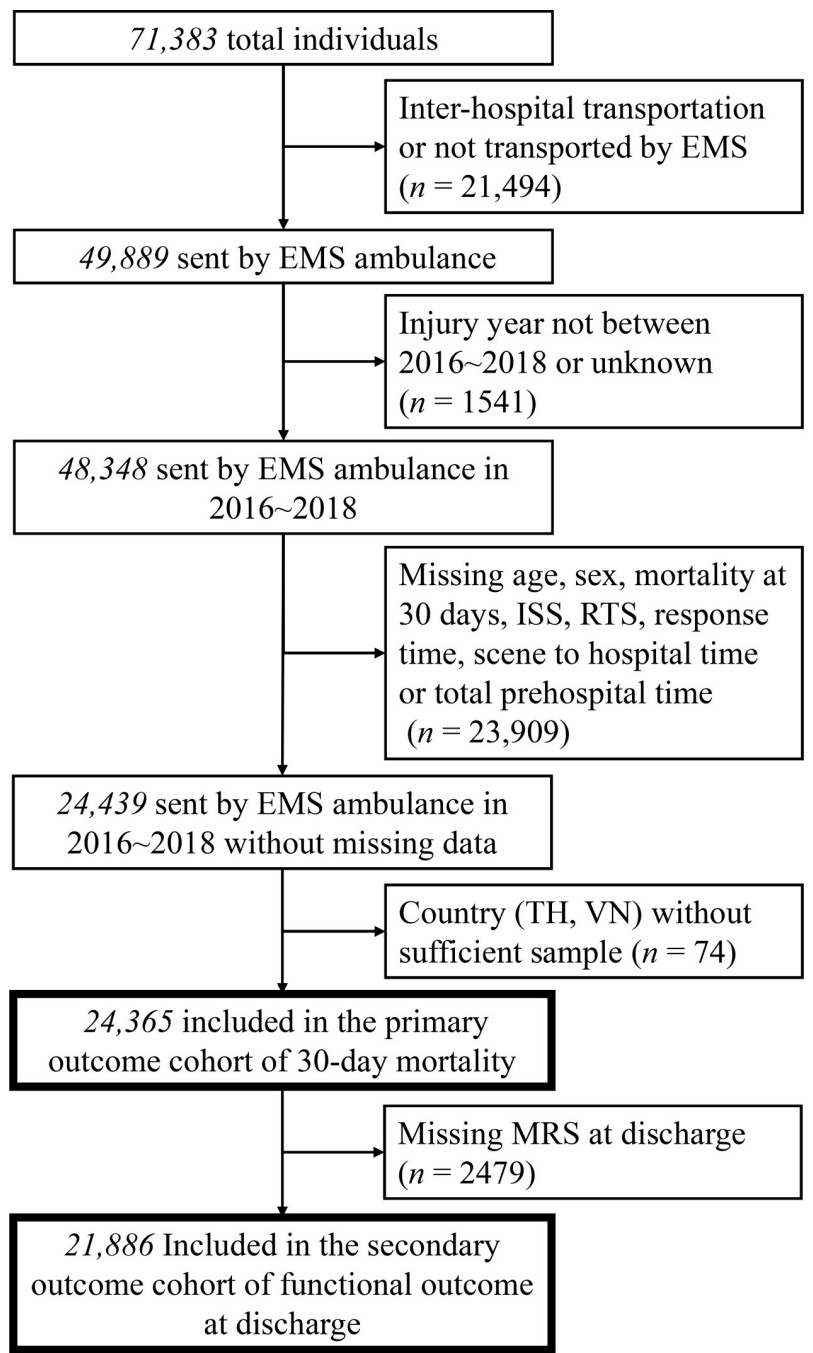

**Fig 1. Flow diagram of patients included in our study.** EMS, emergency medical service; ISS, Injury Severity Score; MRS, Modified Rankin Scale; RTS, Revised Trauma Score; TH, Thailand; VN, Vietnam.

Prehospital times in different subgroups are compared in Table C in S1 Text. RT was significantly shorter in geriatric patients than in non-geriatric patients, while SH was longer in geriatric patients. Median RT and SH were significantly shorter in patients with RTS < 7 (RT: 19.0 versus 21.0 minutes, $p < 0.001$; SH: 20.0 versus 21.0 minutes, $p = 0.015$). Median SH was significantly longer in patients with ISS ≥ 16 (23.0 versus 21.0 minutes, $p < 0.001$). Patients who received a prehospital rescue airway tube had a significantly longer median SH (27.0 versus

**Table 1. Comparison of demographic characteristics of patients included in the study of 30-day mortality.**

| Variable | Total (*n* = 24,365) | Survival at 30 days (*n* = 24,085) | Mortality within 30 days (*n* = 280) | *p*-Value |
|---|---|---|---|---|
| **Country** | | | | <0.001 |
| Japan | 645 (2.6) | 644 (2.7) | 1 (0.4) | |
| Korea | 16,476 (67.6) | 16,371 (68.0) | 105 (37.5) | |
| Malaysia | 5,358 (22.0) | 5,302 (22.0) | 56 (20.0) | |
| Taiwan | 1,886 (7.7) | 1,768 (7.3) | 118 (42.1) | |
| **Age, years** | 45.0 (37.0) | 45.0 (37.0) | 61.0 (34.0) | <0.001 |
| **Male** | 15,498 (63.6) | 15,305 (63.5) | 193 (68.9) | 0.063 |
| **Mechanism of injury** | | | | 0.004 |
| Non-penetrating injury | 23,354 (95.9) | 23,076 (95.8) | 278 (99.3) | |
| Penetrating injury | 1,011 (4.1) | 1,009 (4.2) | 2 (0.7) | |
| **Type of injury** | | | | <0.001 |
| Non-TBI | 16,173 (66.4) | 16,099 (66.8) | 74 (26.4) | |
| Mixed TBI | 3,920 (16.1) | 3,808 (15.8) | 12 (40.0) | |
| Isolated TBI | 4,272 (17.5) | 4,178 (17.3) | 94 (33.6) | |
| **ISS** | | | | <0.001 |
| <16 | 22,727 (93.3) | 22,632 (94.0) | 95 (33.9) | |
| ≥16 | 1,638 (6.7) | 1,453 (6.0) | 185 (66.1) | |
| **RTS** | | | | <0.001 |
| ≥7 | 23,192 (95.2) | 23,121 (96.0) | 71 (25.4) | |
| <7 | 1,173 (4.8) | 964 (4.0) | 209 (74.6) | |
| **Prehospital management** | | | | |
| Supraglottic airway | 11 (0.0) | 6 (0.0) | 5 (3.1) | <0.001 |
| ETT | 7 (0.0) | 5 (0.0) | 2 (0.7) | 0.003 |
| Fluid access (IV, IO) | 1,205 (4.9) | 1,169 (4.9) | 36 (12.9) | <0.001 |
| **Response time, minutes** | 20.0 (27.0) | 20.0 (27.0) | 19.0 (9.8) | 0.020 |
| **Scene to hospital time, minutes** | 21.0 (13.0) | 21.0 (13.0) | 20.0 (12.8) | 0.754 |
| **Total prehospital time, minutes** | 47.0 (28.0) | 47.0 (28.0) | 41.0 (25.3) | 0.006 |

Dichotomous and categorical variables are reported as absolute sample size (percentage), whereas continuous variables are reported as median (IQR).

ETT, endotracheal tube; IO, intraosseous line; ISS, Injury Severity Score; IV, intravenous line; RTS, Revised Trauma Score; TBI, traumatic brain injury.

21.0 minutes, $p = 0.039$) than patients who did not. TPT, RT, and SH were all significantly longer if patients received prehospital IV or IO access. Generally, patients with 30-day mortality had a significantly shorter median TPT (41.0 versus 47.0 minutes, $p = 0.006$), whereas patients with poor functional outcome had a longer median TPT (56.0 versus 48.0 minutes, $p < 0.001$).

## Thirty-day mortality

Multivariable logistic regression with all included patient data ($n = 24,365$) revealed that the RT was not associated with increased risk of 30-day mortality (adjusted odds ratio [aOR] of RT per 10-minute delay: 0.99, 95% CI 0.92–1.06, $p = 0.740$). Long SH may be related to high 30-day mortality, but not at a statistically significant level (aOR of SH per 10-minute delay: 1.08, 95% CI 1.00–1.17, $p = 0.065$). Other variables associated with increased odds for mortality were older age (aOR: 1.03, 95% CI 1.03–1.04, $p < 0.001$), ISS ≥ 16 (aOR: 7.14, 95% CI 5.21–9.78, $p < 0.001$), RTS < 7 (aOR: 28.84, 95% CI 20.61–40.37, $p < 0.001$), rescue airway (aOR: 6.14, 95% CI 1.83–20.60, $p = 0.003$), and establishment of circulatory access (aOR: 1.61, 95% CI 1.04–2.49, $p = 0.033$). TPT was also not significantly associated with increased odds of mortality (aOR of TPT per 10-minute delay: 1.03, 95% CI 0.98–1.09, $p = 0.236$) (Table 2). The

**Table 2. Multivariable logistic regression of 30-day mortality (n = 24,365).**

| Variable | Analysis with RT and SH[†] | | Analysis with TPT[†] | |
|---|---|---|---|---|
| | Adjusted OR (95% CI) | p-Value | Adjusted OR (95% CI) | p-Value |
| **Age** | 1.03 (1.03–1.04) | <0.001 | 1.03 (1.03–1.04) | <0.001 |
| **Sex** | | | | |
| Female | Reference | | Reference | |
| Male | 1.03 (0.76–1.39) | 0.844 | 1.04 (0.77–1.40) | 0.814 |
| **Mechanism of injury** | | | | |
| Non-penetrating | Reference | | Reference | |
| Penetrating | 0.25 (0.05–1.14) | 0.074 | 0.25 (0.06–1.17) | 0.078 |
| **Type of injury** | | | | |
| No TBI | Reference | | Reference | |
| Mixed TBI | 0.96 (0.66–1.40) | 0.843 | 0.98 (0.68–1.43) | 0.918 |
| Isolated TBI | 1.13 (0.78–1.65) | 0.507 | 1.14 (0.78–1.65) | 0.502 |
| **ISS ≥ 16** | 7.14 (5.21–9.78) | <0.001 | 7.21 (5.25–9.88) | <0.001 |
| **RTS < 7** | 28.84 (20.61–40.37) | <0.001 | 28.93 (20.66–40.53) | <0.001 |
| **Prehospital rescue airway**[*] | 6.14 (1.83–20.60) | 0.003 | 6.13 (1.85–20.33) | 0.003 |
| **Prehospital IV/IO access** | 1.61 (1.04–2.49) | 0.033 | 1.60 (1.03–2.47) | 0.035 |
| **RT (per 10 min)** | 0.99 (0.92–1.06) | 0.740 | NA | NA |
| **SH (per 10 min)** | 1.08 (1.00–1.17) | 0.065 | NA | NA |
| **TPT (per 10 min)** | NA | NA | 1.03 (0.98–1.09) | 0.236 |

[†]TPT was put in the multivariable logistic regression separately from RT and SH due to their strong collinearity.

[*]Rescue airway includes prehospital supraglottic airway and endotracheal tube insertion.

CI, confidence interval; IO, intraosseous line; ISS, Injury Severity Score; IV, intravenous line; NA, not applicable; OR, odds ratio; RTS, Revised Trauma Score; TBI, traumatic brain injury; TPT, total prehospital time.

AUROC of the multiple logistic regression model for association of RT and SH and 30-day mortality was 0.95 (95% CI 0.94–0.97), and the Hosmer–Lemeshow test showed inadequate fit ($\chi^2 = 20.70$, $p = 0.008$). The AUROC of the multiple logistic regression model for association of TPT and 30-day mortality was 0.95 (95% CI 0.94–0.97), and the Hosmer–Lemeshow test showed inadequate fit ($\chi^2 = 19.98$, $p = 0.010$). On subgroup analysis, TPT was not associated with 30-day mortality in all subgroups analyzed, including subgroups based on country, age (geriatric/non-geriatric), sex, TBI/non-TBI, penetrating/non-penetrating injury, and different injury severities (Fig 2).

## Functional outcome at discharge

A total of 21,886 patients were included in the study of functional outcome at discharge. Multivariable logistic regression showed significantly increased odds of poor functional outcome with increased RT or SH. The aORs of RT and SH per 10-minute delay were 1.06 (95% CI 1.04–1.08, $p < 0.001$) and 1.05 (95% CI 1.01–1.08, $p = 0.007$), respectively. A 10-minute delay in TPT was also related to 6% increased odds of poor functional outcome (aOR: 1.06, 95% CI 1.04–1.08, $p < 0.001$) (Table 3). The AUROC of the multiple logistic regression model for association of RT and SH and functional outcome at discharge was 0.77 (95% CI 0.76–0.78), and the Hosmer–Lemeshow test showed inadequate fit ($\chi^2 = 37.89$, $p < 0.001$). The AUROC of the multiple logistic regression model for association of TPT and functional outcome at discharge was 0.77 (95% CI 0.76–0.78), and the Hosmer–Lemeshow test showed inadequate fit ($\chi^2 = 36.55$, $p < 0.001$). The results of the subgroup analysis and forest plot are presented in Fig 3.

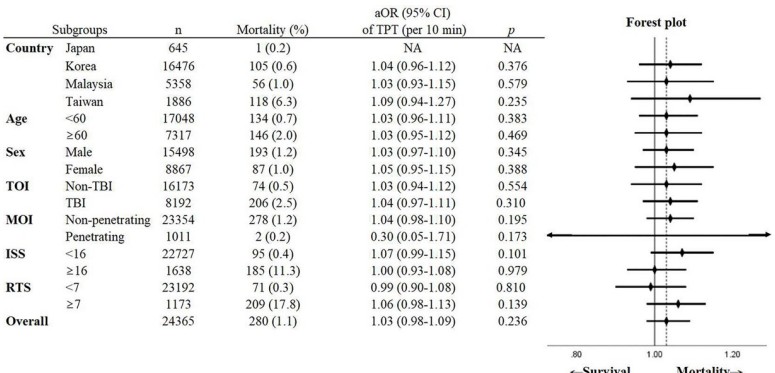

**Fig 2. Subgroup analysis of the association between TPT and 30-day mortality in different subgroups.** *Co-variables used in the logistic regression included prehospital time intervals, age, sex, mechanism of injury, type of injury, Injury Severity Score, Revised Trauma Score, prehospital rescue airway, and prehospital intravenous or intraosseous line, except the variable of the subgroup. All variables were included in the model using a forced entry method. aOR, adjusted odds ratio; CI, confidence interval; ISS, Injury Severity Score; NA, not available; MOI, mechanism of injury; RTS, Revised Trauma Score; TBI, traumatic brain injury; TOI, type of injury; TPT, total prehospital time.

Increased TPT was associated with an increased risk of poor functional outcome in both geriatric and non-geriatric patients, male and female patients, and patients with and without TBI. TPT was not significantly related to functional outcome in Japan and Malaysia and in patients

**Table 3. Multivariable logistic regression of poor functional outcome (*n* = 21,886).**

| Variable | Analysis with RT and SH† | | Analysis with TPT† | |
|---|---|---|---|---|
| | Adjusted OR (95% CI) | *p*-Value | Adjusted OR (95% CI) | *p*-Value |
| **Age** | 1.02 (1.02–1.03) | <0.001 | 1.02 (1.02–1.03) | <0.001 |
| **Sex** | | | | |
| Female | Reference | | Reference | |
| Male | 0.93 (0.82–1.05) | 0.215 | 0.93 (0.82–1.04) | 0.208 |
| **Mechanism of injury** | | | | |
| Non-penetrating | Reference | | Reference | |
| Penetrating | 0.15 (0.07–0.29) | <0.001 | 0.15 (0.07–0.29) | <0.001 |
| **Type of injury** | | | | |
| No TBI | Reference | | Reference | |
| Mixed TBI | 0.79 (0.68–0.93) | 0.003 | 0.79 (0.68–0.92) | 0.003 |
| Isolated TBI | 0.44 (0.37–0.53) | <0.001 | 0.44 (0.37–0.53) | <0.001 |
| **ISS ≥ 16** | 4.77 (4.03–5.66) | <0.001 | 4.73 (3.99–5.61) | <0.001 |
| **RTS < 7** | 7.74 (6.38–9.38) | <0.001 | 7.79 (6.42–9.44) | <0.001 |
| **Prehospital rescue airway*** | 1.88 (0.59–6.04) | 0.288 | 1.88 (0.59–6.01) | 0.290 |
| **Prehospital IV/IO access** | 3.63 (3.01–4.38) | <0.001 | 3.61 (2.99–4.35) | <0.001 |
| **RT (per 10 min)** | 1.06 (1.04–1.08) | <0.001 | NA | NA |
| **SH (per 10 min)** | 1.05 (1.01–1.08) | 0.007 | NA | NA |
| **TPT (per 10 min)** | NA | NA | 1.06 (1.04–1.08) | <0.001 |

†TPT was put in the multivariable logistic regression separately from RT and SH due to their strong collinearity.

*Rescue airway includes prehospital supraglottic airway and endotracheal tube insertion.

CI, confidence interval; IO, intraosseous line; ISS, Injury Severity Score; IV, intravenous line; NA, not applicable; OR, odds ratio; RTS, Revised Trauma Score; TBI, traumatic brain injury; TPT, total prehospital time.

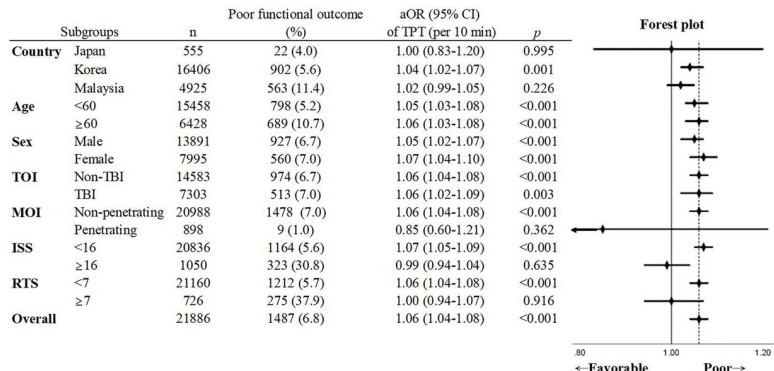

**Fig 3. Subgroup analysis of the association between TPT and poor functional outcome at discharge in different subgroups.** *Co-variables used in the multivariable logistic regression included prehospital time intervals, age, sex, mechanism of injury, type of injury, Injury Severity Score, Revised Trauma Score, prehospital rescue airway, and prehospital intravenous or intraosseous line, except the variable of the subgroup. All variables were included in the model using a forced entry method. aOR, adjusted odds ratio; CI, confidence interval; ISS, Injury Severity Score; MOI, mechanism of injury; RTS, Revised Trauma Score; TBI, traumatic brain injury; TOI, type of injury; TPT, total prehospital time.

with penetrating injury, ISS ≥ 16, and RTS < 7. The restricted cubic spline regression of prehospital time and predicted possibility of favorable functional outcome is depicted in Fig 4. It reveals a nearly linear decrease in predicted possibility of favorable functional outcome as prehospital time increased. Under the receiver operating characteristic curve, a TPT of 49.5 minutes had a maximum Youden Index of 1.10, which indicates that TPT equal to or above 50 minutes best predicts poor functional outcome (Fig A in S1 Text).

## Potential factors influencing TPT

Many factors showed a low correlation with TPT in the linear regression model. Age 60 years and above, male sex, and establishment of prehospital IV or IO were significantly related to long TPT. In contrast, penetrating injury, TBI, and RTS < 7 were associated with short TPT. Prehospital rescue airway establishment and ISS ≥ 16 did not have a significant influence on TPT (Table D in S1 Text).

## Discussion

In this cross-national, multi-center, large-scale retrospective cohort study of the populations in the countries studied from January 1, 2016, to November 30, 2018, we found no association between prehospital time and 30-day mortality in trauma patients. However, increased TPT, RT, or SH may be associated with increased risk of poor functional outcome. To the best of our knowledge, this is the first study to adopt functional outcome as an outcome measurement for examining the impact of prehospital time on injured patients. Functional outcome, which is an index of neurological status, may predict quality of life and the ability to return to normal life and work. Therefore, it is also an important index of outcome, and achieving a favorable functional outcome should be a priority in patient care.

This study has some strengths. First, our study involved the Asian population, which has not been widely investigated in previous studies of the effects of prehospital time on injury outcomes. Moreover, our study included different countries with different EMS systems and enrolled more than 20,000 patients. Therefore, this large cohort may reflect real-world conditions. Second, our study included as many confounders in the multivariable logistic regression

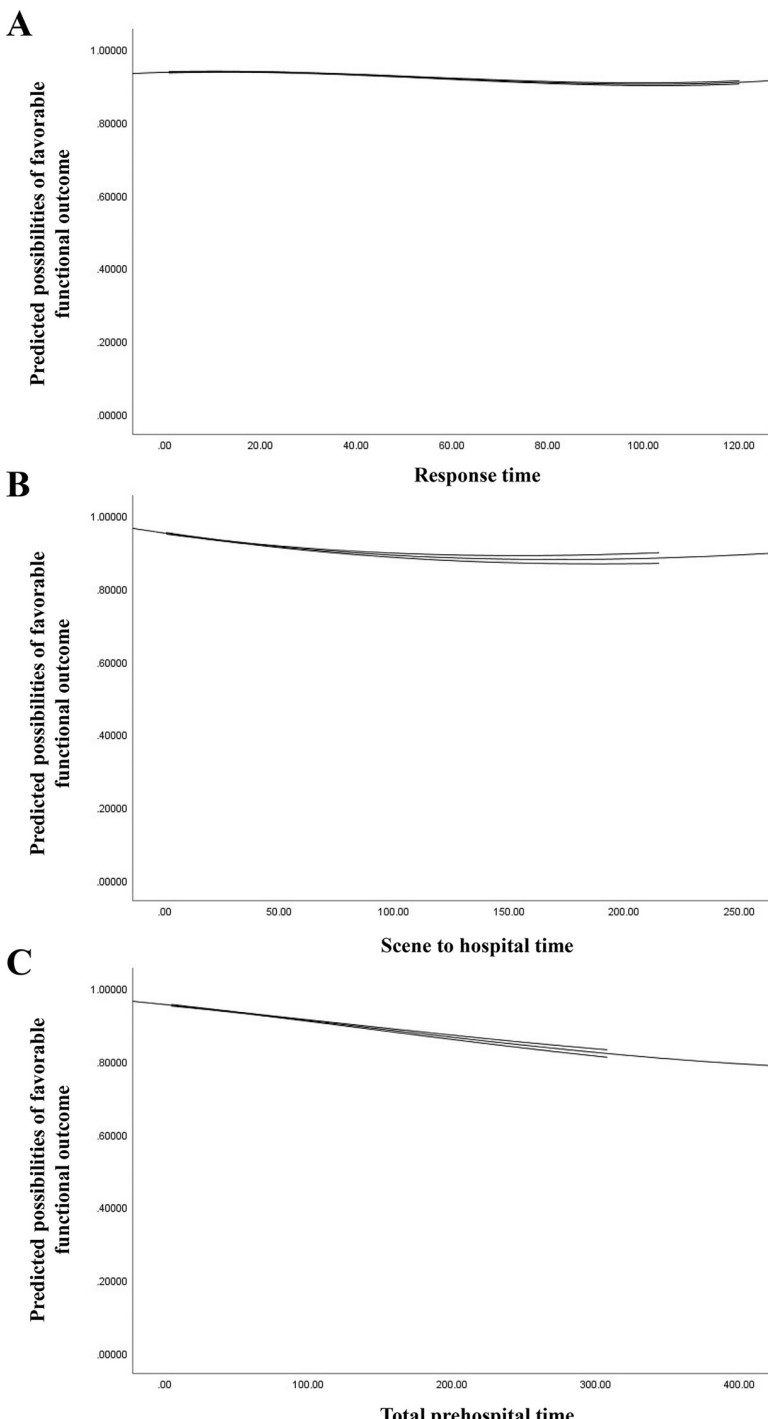

**Fig 4. Restricted cubic spline regression of favorable functional outcome at discharge.** Response time (A). Scene to hospital time (B). Total prehospital time (C).

as possible, including basic demographics, mechanism and type of injury, an injury severity index with clinical significance, and prehospital management. Many previous studies did not include these variables due to limited data validity and may have presented biased results. Third, our study used the timing of injury instead of the timing of calling an EMS as the

starting point in the calculation of prehospital time, to exclude the potential influence of the time between injury and EMS call. The injury time is the core variable in the PATOS registry. The data were usually obtained from caller or patient information. Otherwise, it would be an estimation of the emergency medicine technician (EMT) at the scene. In the outcome measurement, we used 30-day mortality as the outcome index instead of in-hospital discharge, which was commonly used in previous studies. Thirty-day survival status is considered the standard follow-up outcome for trauma patients [26,37].

Previous studies mostly reported no significant association between long prehospital time and poor outcome and refuted the "golden hour" concept [7,38]. Those studies focused on mortality. A prospective study of the Resuscitation Outcomes Consortium epidemiologic out-of-hospital trauma registry in the United States conducted by Newgard et al. [7] presented a robust opposite argument; our study yielded similar results in terms of mortality. However, using functional outcome as an additional outcome measurement, our study showed that patients who experienced transportation delays were likely to have poorer functional outcomes. Therefore, we support the concept of the "golden hour" for trauma patients, emphasizing rapid transportation of injured patients so that they receive timely definite care. Aside from the different outcome measurements used in our study, our study was different from that of Newgard et al. [7] in many ways. The population of included individuals was different. Only adult patients with unstable hemodynamic features were enrolled in their study. Since the study group had specific inclusion criteria, the number of individuals included was limited. In addition, the median prehospital time was shorter in their study setting, due to a different EMS system and catchment area from those in our study.

Several studies have reported increased mortality with delayed prehospital time in patients with penetrating injuries, TBI, or major trauma [10–12,39–41]. In the subgroup analysis of our study, we did not find a specific injury subgroup for which longer TPT was associated with increased risk of 30-day mortality. However, subgroup analysis mostly shared the common limitation of a smaller sample size, which may influence the accuracy of the results. Subgroup analysis of functional outcome was also performed in our study, which supported the result of poor outcome with increased TPT. Most subgroups showed significantly poorer functional outcome with increased TPT, except for those with few participants. The findings of this subgroup analysis strengthened the result that increased TPT is associated with poorer functional outcome.

Our study is based on the PATOS registry data from January 1, 2016, to November 30, 2018. Although there have been significant advances in resuscitation interventions in the prehospital setting, and these might be observed in some major cities, they are not part of daily EMS practices across Asia. Advanced interventions for trauma patients in the prehospital setting will theoretically prolong the prehospital time and are likely to impair the patient outcome, according to our findings. The findings of our study emphasize that EMS should transport injured patients rapidly to a medical facility to ensure favorable functional outcomes. Rapid transportation, rapid response of the EMS team, protocol-based on-scene management of injured patients, and fast decision-making in transporting the patients to appropriate medical centers with dedicated trauma care systems are required. Whether the EMS should make an effort to stabilize or treat patients on scene, i.e., choosing between "load and go" and "stay and play," is also a difficult issue. In our study, we found that, with EMS, establishing circulatory access was associated with increased prehospital time. Patients who were placed on prehospital rescue airway devices may have faced transportation delay, even though this factor did not show a significant effect in the linear regression model. We also reported the surprising fact that patients who received prehospital interventions such as establishing a rescue airway or circulatory access had increased odds of 30-day mortality and poor functional outcomes.

However, this result may be confounded by many uncontrolled factors. The quality and professionalism of each EMS or its members was not known. A retrospective cohort study conducted in Taipei, Taiwan, suggested that prehospital management may improve outcomes for patients with out-of-hospital cardiac arrest only if the quality of the EMS system is assessed as high [42]. Therefore, whether managing the injured patient out of the hospital is beneficial or harmful requires further evidence. Injury severity in another factor that may influence prehospital time. We reported a short TPT for severely injured patients. This finding was in conflict with that of McCoy et al., who suggested that patients with high injury severity may have long prehospital time [11]. Certainly, injury severity may influence the EMS staff's judgement on whether the patient should be sent to the hospital rapidly or be resuscitated at the scene, and on the level of the destination medical center.

It may be reasonable to more specifically investigate patients with major trauma (ISS ≥ 16) because they are more likely to be affected by prehospital time factors. Although it would be more specific to investigate the impact of prehospital time in major trauma patients, we would lose power in the statistical analysis due to inadequate sample size. The results of the subgroup analysis of major trauma patients (ISS ≥ 16) did not show association between prehospital time and outcome. It is also worth noting that the definition of major trauma is based on the ISS, which is calculated at hospital discharge, and not in the field by the EMTs. Since our study focused on prehospital time and prehospital care for patients with trauma, it is reasonable and actually more applicable if we take all trauma patients into consideration, because the results could be used to simply remind the EMTs at the scene to scoop and run, and speed up the transportation of all trauma patients to avoid the risk of poor functional outcome. Another concern was the potential bias caused by variability in the time measurements of different countries. The tools for time measurement indeed varied from site to site. In countries such as Taiwan, Japan, and Korea, centralized timers on electric medical records and ambulance run sheets were used to ensure accuracy. However, some PATOS sites in Malaysia were not equipped with a centralized timer in their measurement and records of ambulance run sheets; thus, random errors could not be totally excluded. We think this kind of random error could be reasonably minimized by using a large sample size. Besides, subgroup analysis showed that longer TPT was not associated with increased risk of 30-day mortality and presented the same pattern of increased risk of poor functional outcome in each country despite the potential difference in time measurements. We further examined the sensitivity of the findings of the logistic regression to mortality within different time periods by performing a sensitivity analysis on different outcomes: mortality within 24 hours, 14 days, 30 days, or 60 days. The results consistently supported that longer prehospital time may not be associated with increased risk of mortality. Also, shorter-term mortality outcomes, such as within 24 hours, rarely occurred in the final analyzed population. In fact, only 92 (0.4%) patients died within 24 hours out of 24,365 enrollees. We think the case number has inadequate power to test the hypothesis.

Our study has some limitations. There is an inherent bias in retrospective cohort studies, mainly due to missing data. Some countries did not record some important variables and had to be excluded from the analysis. For example, MRS at discharge was not recorded in Taiwan (Table E in S1 Text). After enrolling those patients with all valid data and excluding unreasonable data and outliers, the final dataset may have been affected by selection bias. Most of the patients (14,678 patients) excluded due to missing data had one of the prehospital time records that we aimed to investigate missing, instead of missing records of age, ISS, or other physiological variables. Therefore, we think that it would be inappropriate to impute data of prehospital times, whereas it would be of less value to impute physiological variables other than prehospital times. We further compared the basic characteristics of the included sample and the sample of individuals excluded due to missing data and found differences in patients' age, injury

severity, RT, SH, and outcome (Table F in S1 Text). However, the main differences between the 2 samples were the outcome variables. As the EMTs who recorded the prehospital time variables may not be able to predict the patients' outcome, much of this missing data might be reasonably seen as missing at random. Although we excluded half of the sample in our analysis, we still investigated more than 20,000 patients in this cross-national, multi-center cohort study, which should still be a valuable and informative contribution to the knowledge gap in this area. Nevertheless, we certainly agree that our analysis is susceptible to potential bias resulting from the differences in basic characteristics of the included and excluded sample and missing data inherent to the retrospective design.

Another limitation is that studies from registries may be subjected to common bias caused by being unable to account for patients who were sent to the participating hospitals. The number of patients lost to follow-up was also unknown. Furthermore, the included countries had relatively large differences in terms of patient number, mortality, and injury severity, which were not matched in our study (Table E in S1 Text). Moreover, we included many variables in the logistic regression model, but many unknown factors can influence mortality, such as environmental factors, bystander management of the patient, and the quality of each EMS team. In-hospital variables such as complications, procedures, and blood and crystalloid resuscitation volumes were not able to be included in our analysis either due to missing data or nonrecording of the variable in the registry. We were unable to match all the confounders to determine the direct effect of prehospital time on outcome. On the other hand, although in-hospital variables will affect patients' outcomes, it is less likely that they are potential confounders of the association between prehospital time and outcome because in-hospital treatment is theoretically dependent on a patient's condition and is not associated with the prehospital timeliness. The population, total EMS catchment area, EMS system, and trauma care education program and practices also differed among the countries included in this study [19]. This heterogeneity may cause potential confounding in the result and difficulties in further application of our results to different countries.

Data from Taiwan included ISS based on manually converted AIS 2008 codes from AIS 1998 codes, whereas ISS from other countries was based on AIS 2008. For the sake of statistical validity, we also performed a logistic model for mortality excluding data from Taiwan, and the association between TPT and both outcomes remained unchanged. Data from Taiwan did not contain MRS at hospital discharge, and thus were not included in the analysis of functional outcome. Another essential limitation is that there may be potential differences between countries and hospitals that may require hierarchical nested models that account for patient clustering. Using models accounting for clustering is another statically valid method to examine the association, and could change the confidence intervals for the same parameter estimates [43]. However, we performed many subgroup analyses, and found that longer TPT was not associated with increased risk of 30-day mortality and consistently presented the same pattern of in increased risk of poor functional outcome in each country. The difference between each included country seemed to be small compared to the pooled data of the included countries. Furthermore, hospitals enrolled in the PATOS registry were all teaching hospitals capable of trauma resuscitation. In light of the above reasons, we assumed the difference among countries and hospitals may be small. Besides, although the discrimination of our multiple logistic regression model for mortality prediction was satisfactory, we observed significant inadequate fit using the Hosmer–Lemeshow statistic. The large sample size, which decreases the power of any goodness of fit test, may explain the poor calibration [44,45]. We further depicted plots of predicted to observed outcome rates of TPT per 10 minutes to permit visual inspection of model calibration in Fig B in S1 Text. Except for prehospital time intervals with 0% observed outcome rate, the model fitness was relatively satisfactory by visual inspection. Finally, detailed

subgroup analysis and data on time to definite care (surgery or trans-arterial embolization) were limited due to the sample size and number of valid data.

Despite subgroup analysis showing the same pattern in the different studied countries that longer prehospital time may be associated with increased risk of poor functional outcome, the result only showed statistical significance in Korea. EMS teams from different countries and areas should make an effort to study the influence of prehospital time in their own systems and develop suitable guidelines based on their prehospital care setting and trauma care capacity. Also, further robust studies to verify the results of our study may still be required.

## Conclusions

Although there was no significant association between prehospital time and 30-day mortality in trauma patients, our study supported rapid transportation for all injured patients because longer prehospital time may be associated with an increased risk of poor functional outcomes; the odds of poor functional outcome increase by 6% with every 10-minute delay in TPT, and TPT longer than 50 minutes can predict poor outcome. We support the concept of the "golden hour" for trauma patients during prehospital care in the countries studied. Further protocol setting in prehospital management by each EMS team and verification of the effect are required. Each EMS team should also develop suitable guidelines based on their prehospital care setting and trauma care capacity within optimized prehospital time.

## Supporting information

**S1 Checklist. STROBE checklist.**
(DOCX)

**S1 Text.** Tables A–F and Figs A and B.
(DOCX)

## Acknowledgments

The authors acknowledge all participating PATOS sites for excellent collaboration, as well as the data quality assurance of the PATOS coordination center at Seoul National University Hospital, to improve prehospital and in-hospital care for trauma patients in Asia. The authors also like to thank the staff, Dr. Chin-Hao Chang, of the National Taiwan University Hospital Statistical Consulting Unit for help in statistical consultation.

## Author Contributions

**Conceptualization:** Wen-Chu Chiang.

**Data curation:** Sang Do Shin, Jen-Tang Sun, Sabariah Faizah Jamaluddin, Hideharu Tanaka, Kyoung Jun Song, Kentaro Kajino, Akio Kimura, Wen-Chu Chiang.

**Formal analysis:** Chi-Hsin Chen, Wen-Chu Chiang.

**Investigation:** Chi-Hsin Chen, Sang Do Shin, Jen-Tang Sun, Sabariah Faizah Jamaluddin, Hideharu Tanaka, Kyoung Jun Song, Kentaro Kajino, Akio Kimura, Edward Pei-Chuan Huang, Ming-Ju Hsieh, Matthew Huei-Ming Ma, Wen-Chu Chiang.

**Methodology:** Chi-Hsin Chen, Wen-Chu Chiang.

**Project administration:** Sang Do Shin, Jen-Tang Sun, Wen-Chu Chiang.

**Resources:** Sang Do Shin, Ming-Ju Hsieh, Matthew Huei-Ming Ma, Wen-Chu Chiang.

**Supervision:** Jen-Tang Sun, Edward Pei-Chuan Huang, Ming-Ju Hsieh, Matthew Huei-Ming Ma, Wen-Chu Chiang.

**Validation:** Chi-Hsin Chen, Wen-Chu Chiang.

**Writing – original draft:** Chi-Hsin Chen.

**Writing – review & editing:** Wen-Chu Chiang.

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
