## [Editor Report · Decision Letter 0]

3 Jun 2020

Dear Dr WEN-CHU, 

Thank you for submitting your manuscript entitled "Association between prehospital time and outcome of trauma patients in Asia: Does the “golden hour” matter?" for consideration by PLOS Medicine.

Your manuscript has now been evaluated by the PLOS Medicine editorial staff as well as by an academic editor with relevant expertise and I am writing to let you know that we would like to send your submission out for external peer review.

Kind regards,

Artur Arikainen,

Associate Editor

PLOS Medicine

---

## [Decision Letter · Decision Letter 1]

23 Jun 2020

Dear Dr. WEN-CHU,

Thank you very much for submitting your manuscript "Association between prehospital time and outcome of trauma patients in Asia: Does the “golden hour” matter?" (PMEDICINE-D-20-02427R1) for consideration at PLOS Medicine. 

[LINK]

In light of these reviews, I am afraid that we will not be able to accept the manuscript for publication in the journal in its current form, but we would like to consider a revised version that addresses the reviewers' and editors' comments. Obviously we cannot make any decision about publication until we have seen the revised manuscript and your response, and we plan to seek re-review by one or more of the reviewers. 

We expect to receive your revised manuscript by Jul 14 2020 11:59PM. Please email us (plosmedicine@plos.org) if you have any questions or concerns.

We look forward to receiving your revised manuscript. 

Sincerely,

Emma Veitch, PhD

PLOS Medicine

On behalf of Clare Stone, PhD, Acting Chief Editor,

PLOS Medicine

plosmedicine.org

*We'd suggest revising the title according to PLOS Medicine's style - after the study objective/question (in the first phrase of the title), we'd normally have a colon and then the study design, eg here "xxyy: retrospective cohort"). 

*Please structure the abstract using the PLOS Medicine headings (Background, Methods and Findings, Conclusions) - "Methods and Findings" is a single subsection heading. 

*In the last sentence of the Abstract Methods and Findings section, please include a brief summary of any of the key limitation(s) of the study's methodology.

*Currently in the abstract conclusions, the paper states "Long prehospital time ... but significantly affected the functional outcome of injured patients". However, this seems to go beyond what the retrospective cohort design is capable of concluding, ie this makes an assertion about causality rather than stating the findings as associations or links only. We'd suggest rephrasing this (and elsewhere in the paper) where a causal inference is made on the basis of observational data.

*Ideally please reformat the in-text reference citations to use sequential numerals in square brackets (eg, [1], [2], etc).

*Please clarify in the paper if the analysis reported here corresponds to one that was laid out in a prospectively-developed protocol or analysis plan? Please state this (either way) early in the Methods section.

*In the Methods section where the authors say multivariate, do they mean multivariable? 

*In the Discussion section, more care is needed in a number of places with regard to causal inference - as noted above. For example, "However, increased TPT, RT, or SH may significantly increase the risk of poor functional outcomes" - this would be better stated as "associated with increased risk of...". Similarly "our study suggested rapid transportation for all injured patients because increased prehospital time may lead to poor functional outcomes" could be rephrased more precisely. 

Comments from the reviewers:

Reviewer #1: Thank you very much for the opportunity to review a manuscript titled 'Association between prehospital time and outcome of trauma patients in Asia: Does the "golden hour" matter?' This study investigated the association between prehospital time factors and survival and functional outcomes using data from several countries in Asia. The authors found an increasing risk for adverse functional outcome as prehospital time increases. The methods used in the study appear to be appropriate and the results seem to be reasonable. However, I have a concern re characteristics of patients included. 93.3% of the patients had ISS<16, which is deemed as non-major trauma. Therefore, outcomes of most participants are less likely to be affected by prehospital time factors. I feel that prehospital time factors may be more important for major trauma patients. It would be useful if you could repeat this analysis by including only major trauma patients, but I understand sample size will significantly decrease by doing so and you will lose power. My other minor concerns are described below.

The 2nd last sentence in a paragraph with a header of 'Study design and setting', Malaysia appears twice in the sentence.

Do all participating countries use the same version of the AIS to compute ISS? If not, please address the issue.

In 'Statistical analysis': 'Continuous dependent variables were compared using non-parametric ANOVA and Mann-Whitney U test. Categorical and nominal dependent variables were compared using the chi-square and Pearson Chi-square test or Fisher's exact test.'

I may be misunderstanding, but your dependent variables are 'mortality' and 'functional outcome', aren't they? I presume you meant to say 'independent variable'? Apology if I misunderstood.

'Continuous dependent variables were compared using non-parametric ANOVA and Mann-Whitney U test.' I think you tested using either methods. So, it should be 'ANOVA or Mann-Whitney'

'Categorical and nominal dependent variables were compared using the chi-square and Pearson Chi-square test or Fisher's exact test.' -> What is the difference between 'chi-square test' and 'Pearson Chi-square test'? Did you use other type of chi-square test than Pearson chi-square?

'To determine the potential factors that may influence scene to hospital time, variables that had a p <0.10 on the Mann-Whitney U test were selected for a multivariate linear regression analysis using the forced entry method.'

It is not clear why you derived a linear regression model because the 2 outcome variables are dichotomous (mortality and functional outcome). Please explain why 'scene to hospital time', not total prehospital time or scene time, was used as a dependent variable in the linear regression.

'The restricted cubic spline regression of prehospital time and predicted possibilities of favourable functional outcome are depicted in Figure 4.' in Functional outcome at discharge in the results-> This is not described in the Methods. 

The authors reported factors associated with total prehospital time, but this is not part of aims. It is ok to analyse the association, but the authors need to present it as an aim and give some rationale to investigate it if possible. Otherwise, I was unsure why the authors derived the linear regression model.

I could not find Table E1, E2, E4. There is no Table E3.

Reviewer #2: The authors have conducted a retrospective cohort study of trauma patients transported to hospitals by emergency medical service (EMS) between 2016 and 2018 using data from the Pan-Asia Trauma Outcomes study (PATOS) database.

Comments:

Is the timeframe of the data (2016-2018) still relevant in today's clinical setting? Has medical equipment available in transit changed since then, for instance?

The authors acknowledge that "... there have been significant advances in resuscitation, airway management, circulatory access, and hemorrhage control as the prehospital care provided by paramedics and other emergency medical technicians".

Whilst briefly mentioned in the limitations, how do the authors expect excluding patients based on missing values might have affected their results? 

Can these patients/values be assumed to be missing at random? Are the remaining patients included in the analysed sample representative of the wider population?

"...age was below 1 year or above 120 years, which were considered unreasonable data and potential outliers."

Can the authors explain why age below 1 was considered an outlier?

Did the authors consider variability in the time measurements they included, and how uncertainty in measurements might impact their results?

The statistical techniques and tests used appear to be appropriate for the data type and research question.

How sensitive are the findings of the logistic regression to the cut off of 30-day mortality? Do 20- or 40- day mortality models show any substantial difference in the results?

Albeit the authors claim that "30-day survival status is considered the standard follow-up outcome for trauma patients."

Did the authors explore and include interaction variables within their regression modelling?

Did the authors consider differences by the four included countries, or just look at the pooled Asia scenario?

Reviewer #3: The research question and topic are certainly of interest. The authors present a multi-county evaluation of Asian trauma patients and the association of prehospital time with mortality and functional status at discharge. 

The background is logical but too long. The authors review several individual prior studies on prehospital time in detail that distracts from focusing on their objectives and hypothesis. This review of the existing literature would be better served in the discussion of the paper. 

The authors clearly outline the objectives. The authors should explicitly state their hypothesis regarding the relationship between prehospital time and outcomes in their population.

The methods require some clarification and I have several concerns as below: 

1. The database used for the study requires further description. Which hospitals submit data to the database? All hospitals, only trauma centers? Is data submission voluntary or required of trauma centers? Do the types of hospitals include vary across countries in the database?

2. Given variation in prehospital systems, the authors should provide a brief description of the EMS systems and capabilities in the studied countries. Are there significant differences between the counties in the EMS systems for the patients studied?

3. It is not clear from the methods if the patients included were only patients transported from the scene, patients transferred between hospitals, or both?

4. The authors exclude nearly half of the eligible patients for missing data. The authors should consider multiple imputation which has been validated in similar large trauma registry databases to avoid systematic bias that may result from excluding all these patients. The authors should also compare patients included and excluded to evaluate for potential bias from excluding these patients

5. Why did the authors no evaluate transport time as one of the prehospital intervals (time from leaving the scene to arriving at the hospital)? This may be an important interval to evaluate as well independent of the total prehospital time. 

6. The authors state a strength of the study was using time of injury rather than time of EMS activation. How was this data known and collected, and how reliable is it?

7. Did the authors consider a shorter-term mortality outcome such as 24 hours? Many factors between prehospital transport and 30 days may occur that obscure the impact of prehospital times on mortality

8. Related to the above point, the authors do not adjust for any in-hospital variables that may confound mortality (e.g. complications, procedures, blood and crystalloid resuscitation volumes). The authors should adjust for these factors if evaluating 30-day mortality

9. Are patients that died before reaching the hospital or died at non-trauma centers or small community hospitals included in the database? I am concerned there is a survival bias in the dataset in which patients that had longer prehospital time may have died before reaching a hospital that includes data, and patients with long prehospital times in the dataset were at low risk of dying, making it look like there's no relationship or even that longer prehospital times were protective against death. 

10. Do the authors know who administered the MRS and whether a structured questioner was used? Studies show poor interrater reliability without a structured questioner. This may be even more important since it has not robustly been validated in the general trauma population (only case-control study cited by the authors). 

11. The authors should use models that account for patient clustering in the different countries and hospitals in the dataset (e.g. hierarchical nested models)

12. The authors should present discrimination and calibration assessments of their outcome models

13. There is a discrepancy for the linear regression models - in the methods it states scene to hospital time, which is transport time, but in the results the authors discuss total prehospital time. Please clarify the prehospital time interval used

The results are well organized. Some of the introductory demographics text can be removed as it is redundant with Table 1. 

The authors present results of a restricted cubic spline regression that was never mentioned in the methods. The authors should include all planned and post-hoc analyses in the methods section. 

Minor corrections - 

"Multivariate" should be changed to "multivariable" regarding the models throughout the text

On the third page of the discussion "scene and play" should be "stay and play" 

While the authors should be commended for evaluating prehospital time in a large previously understudied population, there are several methodologic concerns and some inherent to the database which limits conclusions that can be drawn from this manuscript.

[LINK]

---

## [Decision Letter · Decision Letter 2]

10 Aug 2020

Dear Dr. WEN-CHU,

Thank you very much for re-submitting your manuscript "Association between prehospital time and outcome of trauma patients in Asia: A cross-national, multicenter, retrospective cohort study" (PMEDICINE-D-20-02427R2) for review by PLOS Medicine.

I have discussed the paper with my colleagues and the academic editor and it was also seen again by two reviewers. I am pleased to say that provided the remaining editorial and production issues are dealt with we are planning to accept the paper for publication in the journal.

[LINK]

We look forward to receiving the revised manuscript by Aug 17 2020 11:59PM. 

Sincerely,

Artur Arikainen, 

Associate Editor 

PLOS Medicine

plosmedicine.org

Requests from Editors:

1. Please address the comments of reviewers #1 and #3 below. Specifically, please ensure that you highlight all of the limitations of your study raised by reviewer #3.

2. Title: Please amend to: “Association between prehospital time and outcome of trauma patients in Asia: A crossnational, multicenter, cohort study”

3. Please remove spaces from within citation callouts and place the callout before punctuation, eg. “…mortality annually worldwide [1,2].” Please keep the space between the text and citation itself.

4. Abstract:

a. Line 4, please rephrase the following, as ‘impact’ may imply causation: “…to explore the concept of a “golden hour” on injured patients.”

b. Please include basic patient demographics (median age and range, sex) around line 11.

c. Please also add a breakdown of participants by country.

d. Please state which factors were adjusted for in your odds ratios.

e. Lines 12-13: Please present IQRs as upper and lower value here and in the main Results section.

f. Please quantify all results with p values. Please also include p values and odds ratios for the following statement: “Prehospital time intervals were not associated with 30-day mortality.”

g. Please mention another limitation at the end of the Methods and findings subsection, eg. possible unaccounted for confounding factors.

h. Line 23, please rephrase to: “…but it may be associated with increased risk of poor functional outcomes in injured patients.”

6. Throughout, please avoid referring to “in Asia”, as this would be an overgeneralisation of the scope of your study – instead please refer to “the countries studied” or list the countries by name.

7. Methods:

a. Page 5, Lines 23-24: Please give the exact date ranges covered by your study, including day and month.

b. Page 6, Lines 18-20: Please clarify whether data were fully anonymised at the time they were accessed by the authors, or whether patients provided written informed consent to be included in the study.

c. Page 6, Line 25: Correct to “…data were…”

d. Please clarify which analyses were carried out in response to peer review, where differing from the original study plan.

8. Results:

a. Please present ORs, CIs and p values to the same number of decimal digits throughout, eg. page 11, lines 21 and 24.

b. Page 12, Line 1: Please replace “gender” with “sex”. The terms gender and sex are not interchangeable (as discussed in http://www.who.int/gender/whatisgender/en/ ).

9. Discussion:

a. Page 16, lines 14-15: Please remove this sentence: “We truly appreciate peer reviewers in putting forward some insightful concerns of our study.” Please rephrase the rest of that paragraph to avoid specific references to the peer review process.

b. Page 20, line 4: Please amend to: “…our study support rapid transportation for all injured…”

10. Page 20: Please remove the ‘Conflicts of interest’ section – this is taken from the online submission form instead.

11. Table 1: Please spell out the country names in the variables column, for clarity.

12. When completing the STROBE checklist, please use section and paragraph numbers, rather than page numbers. Please add the following statement, or similar, to the Methods: "This study is reported as per the Strengthening the Reporting of Observational Studies in Epidemiology (STROBE) guideline (S1 Checklist)."

----

Comments from Reviewers:

Reviewer #1: Thank you very much for the opportunity to re-review a manuscript titled 'Association between prehospital time and outcome of trauma patients in Asia: Does the "golden hour" matter?' The authors sufficiently and appropriately responded to all of my questions and the manuscript was amended accordingly. The only concern is that data from Taiwan was included when deriving a model. The authors answered that data from Taiwan included ISS based on AIS 98, whereas ISS from other countries were based on AIS 2008. It is known that ISS based on AIS 2008 is significantly lower than that based on AIS 98. Therefore, it is not statistically valid to derive a model by using data from various countries, which use different version of AIS. I suggest that the authors should derive a logistic model for mortality after excluding data from Taiwan. The author can also drive a model only using data from Taiwan. Other solution could be converting ISS based on AIS 98 to ISS based on AIS2008 if the authors have AIS codes for cases from Taiwan. I hope these comments would help the authors to improve the quality of this manuscript. Thank you very much.

Reviewer #3: The authors should be commended for substantial revisions to address many of the issues raised. 

I have the following remaining concerns: 

1. The authors have included a comparison of included and excluded patients based on missing data. These new data show the excluded patients more often had an ISS>=16 or RTS<7, making them a sicker more severely injured cohort. While I agree with the authors that imputing prehospital times would not be appropriate, imputing age, ISS, and physiologic variables may be to reduce the patients excluded. While I agree the authors still have 20,000 patients to study, the excluded patients appear to by systematically different in some important ways which is irrelevant to the sample size. 

2. From the response, the authors may had misunderstood my concern regarding potential survival bias. They comment on patients who may have died or arrested in the ambulance on the way to the hospital which was a small proportion of patients. I believe the larger issue are the patient who do not appear in the database because they went to a hospital that does not participate in the PATOS database and then died, or died before any EMS arrived. Those are the patients where prolonged prehospital time would potentially be detrimental but won't show up in the database at all. This is a common limitation to registries that collect data from major trauma centers only and should be acknowledged. 

3. The authors may have also misunderstood the issues of using hierarchical models to account for clustering. They note in their comments there was similarly no mortality association in subgroup by country, and a similar trend by county for functional outcome (although it actually shows only Korea had a significant association between prehospital time and functional outcome). Similar subgroup results is not the same issue statistically that can result in biased estimates. Clustering of patients by country in the models accounts for the fact that the patients are not completely independent observations which is an assumption in logistic regression. Patients in similar countries are more likely to have similar outcomes in both mortality and functional outcomes, making them not completely independent. Not accounting for this can result in biased and artificially narrow confidence intervals for the model parameter estimates. Given that the confidence intervals for the effect of prehospital time on functional outcome are close to 1, this may in fact change the main results of the paper and should be considered. See the following: Roudsari et al. Clustered and missing data in the US National Trauma Data Bank: implications for analysis. Inj Prev. 2008;14:96-100 and Roudsari et. Analysis of clustered data in multicentre trauma studies. Injury. 2006;37:614-621. 

4. Related to the above, the authors present the results as a cross-national study with the main finding of poor functional outcome associated with increased prehospital time; however, it appears this was only the case in Korea, while no other associations were significant. I believe the warrants some comment in the discussion. It may be under-powered (particularly in Japan) or may truly be an effect only in the Korean system (Malaysia with 5000 patients and >10% event rate should be reasonable to detect an effect). 

5. The authors are correct the Hosmer-Lemeshow test performs poorly in large sample sizes. The authors should include plots of predicted to observed outcome rates to permit visual inspection of model calibration since the H-L test is of little utility in this circumstance. 

While I do think this study can add to the literature, I believe there are additional steps the authors should take to further limit bias is their analysis. Further, there are several limitations inherent to the database that limit the rigor and conclusions that can be drawn from this study. Given these issues, this study may be better served in a specialty journal focused on trauma or EMS.

[LINK]

---

## [Editor Report · Decision Letter 3]

31 Aug 2020

Dear Dr. Wen-Chu, 

On behalf of my colleagues and the academic editor, Dr. Karim Brohi, I am delighted to inform you that your manuscript entitled "Association between prehospital time and outcome of trauma patients in four Asian countries: A cross-national, multicenter, cohort study" (PMEDICINE-D-20-02427R3) has been accepted for publication in PLOS Medicine. 

PRODUCTION PROCESS

PRESS

PROFILE INFORMATION

Thank you again for submitting the manuscript to PLOS Medicine. We look forward to publishing it. 

Best wishes, 

Artur Arikainen, 

Associate Editor 

PLOS Medicine

plosmedicine.org